# A Generation-based Deductive Method for Math Word Problems

**Yuxuan Hu**[1,2], **Jing Zhang**[1,3]*, **Haoyang Li**[1,3], **Cuiping Li**[1,3], **Hong Chen**[1,3]

[1]School of Information, Renmin University of China, Beijing, China
[2]Key Laboratory of Data Engineering and Knowledge Engineering, MOE, China
[3]Engineering Research Center of Database and Business Intelligence, MOE, China
{huyuxuan1999,zhang-jing,lihaoyang.cs,licuiping,chong}@ruc.edu.cn

## Abstract

Math word problems (MWP) involving advanced operators such as linear equation solver cannot be easily tackled by earlier MWP methods, because the existing generation methods suffer from repeated sub-expression generation and deductive methods are restricted to dealing with binary operations. This paper propose a new multivariate directed acyclic graph (mDAG) as an alternative to the generation methods' binary expression tree or the deductive methods' binary directed acyclic graph. Then to produce the topological ordering of mDAG, we propose a generation-based deductive (GeDe) model, which equips a generation model with a re-encoder to keep the deductive property but avoid the expensive enumeration of the deductive methods. GeDe performs well on math problems with many operators on the widely used benchmarks as well as solving multivariate operators on our own CMWPA benchmark. Our code is available at https://github.com/hyx1999/GeDe

## 1 Introduction

Solving Math Word Problems (MWPs) is the task of answering natural language problems that require mathematical reasoning ability (Bobrow, 1964). To achieve such a skill, researchers have proposed a variety of MWP solvers, each of which seeks to produce a specific logic form that can be used to calculate the answer to the problem.

Deductive methods and generation-based methods are typically the two main approaches used to solve MWPs. Inspired by advances in machine translation, some generation-based methods directly adopt a sequence-to-sequence (seq2seq) model to generate the sequence of the math expression according to the problem (Wang et al., 2017). To further capture the structure of the math expression, some sequence-to-tree (seq2tree) methods (Xie and Sun, 2019) adopt a tree decoder to

---

*Corresponding author.

generate the binary expression tree, where each node denotes an operator or a quantity. These generation-based methods, however, suffer from a fatal flaw in that they require repeated generation of the same sub-expression (or sub-tree), which makes them inefficient. For example, in Figure 1 (a), the sub-expression $(94 - 35 \times 2) \div (4 - 2)$ is generated four times. Humans, on the other hand, can represent repeated sub-expressions with an intermediate quantity that can be naturally reused in the following computation process.

Deductive approaches (Cao et al., 2021; Jie et al., 2022) are suggested to address the aforementioned reuse issue. Specifically, deductive methods convert the math expression into a binary Directed Acyclic Graph (bDAG), where each node represents an operation that consists of a binary operator and two input quantities. The calculation result of an operation is represented by a new intermediate quantity. Then, these methods need to generate a topological ordering, *i.e.*, an operation sequence, of the bDAG. By doing this, subsequent operations can easily reuse the previously generated intermediate quantities. As shown in Figure 1 (b), quantity $q_3$ represents the sub-expression $(94-2\times35)\div(4-2)$, which is then reused by two subsequent operations denoted by quantity $q_4$ and $q_8$. When the operation sequence is inferred, these operations are computed consecutively to produce the final answer. Beyond the ability to reuse the intermediate quantity, deductive methods are more interpretable because the step-by-step generation of operations helps people understand how the reasoning works. To generate the operation at each reasoning step, existing deductive methods follow an "enumerate-then-classify" procedure. To be more precise, they create a collection of candidate operations by listing every possible combination of the quantities and operators, and then they use a classifier to choose the operation that has the highest probability, which can be viewed as a greedy search strategy.

**Question:**
*There are several chickens and rabbits in a cage. Inside, we observe **94** feet and **35** heads. A chicken has **1** head and **2** feet. A rabbit has **1** head and **4** feet. The number of rabbits and chickens are denoted by **x** and **y**, respectively. Tell me the value of x × x + y ×y.*

**Answer:** 673    **Mathematical Expression:**
**((94-2×35)÷(4-2))**×**((94-2×35)÷(4-2))** +(35- 1× **((94-2×35)÷(4-2))** ÷1)×(35- 1× **((94-2×35)÷(4-2))** ÷1)

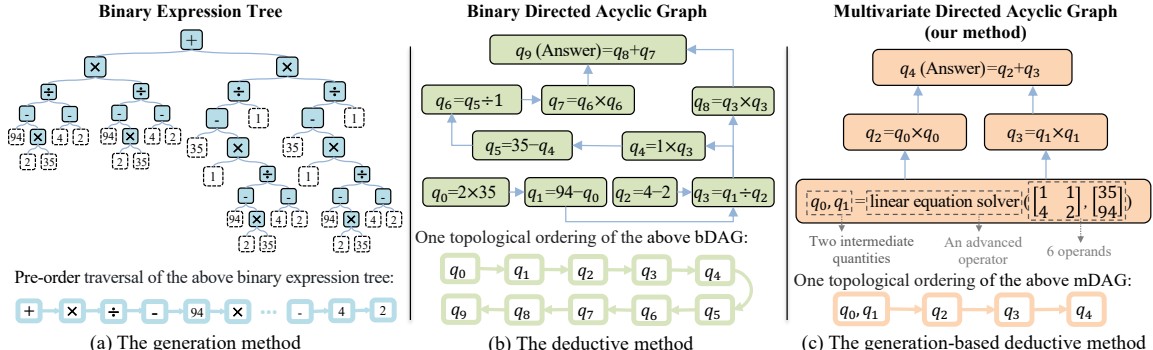

Figure 1: Illustration of a MWP example with a natural language input problem and a corresponding mathematical expression output that can be used to calculate the answer. The repeated sub-expression is underlined in red. In order to get the answer, three methods are presented: (a) a seq2seq or seq2tree generation method to generate an expression sequence or a binary expression tree; (b) a deductive method to reason out a topological ordering of the bDAG; and (c) the proposed generation-based deductive method to generate a topological ordering of the mDAG.

One obvious limitation of the aforementioned approaches is that they only take into account the basic binary operators such as $+, -, \times, \div$. Although binary operators are the most fundamental in mathematics, there are some templated problems, such as solving linear equations, finding the extreme values of quadratic equations, and even integrating a function, that can be solved by existing advanced operators. Thus, we can abstract an **advanced operator** to tackle each templated problem. With these advanced operators, we can inject prior mathematical knowledge to reduce the difficulty of solving MWPs. However, problems requiring advanced operators are difficult to tackle using earlier MWP methods: generation-based methods inherently suffer from the reuse issue; deductive methods are limited by the assumption of binary operations.

To address this issue, we first define a multivariate Directed Acyclic Graph (mDAG) with each node involving a multivariate operation that consists of a basic or advanced operator and multiple input quantities. Compared to basic binary operators, advanced operators can receive multiple quantities and return multiple output quantities. For example, in Figure 1 (c), a linear equation solver requires 6 quantities $(1, 1, 4, 2, 35, 94)$ and returns 2 intermediate quantities $(q_0, q_1)$. Then, similar to the bDAG, we use the topological ordering of the mDAG to obtain a sequence of multivariate operations. To generate such a sequence, we propose GeDe, a Generation-based Deductive approach.

Compared to generation-based techniques, GeDe has the deductive property that enables the reuse of intermediate quantities. Compared to deductive methods, GeDe employs a generation model to generate each multivariate operation, which avoids the need to enumerate a large number of possible multivariate operations.

In order to achieve this generative-based deductive capacity, we equip a generation model with a re-encoding strategy that jointly encodes the problem and intermediate quantities at each step of reasoning, yielding embeddings of the intermediate quantities that could be reused in the subsequent steps. In addition, we switch from the traditional greedy or beam search to a hierarchical beam search strategy, which is well suited to the equation generation requirement.

**Contributions.** (1) By extending bDAG to mDAG, we can directly address complex mathematical problems using pre-defined advanced operators. (2) We propose GeDe, a generation-based deductive model that keeps the deductive property while avoiding the high cost of enumeration. GeDe equips a generation model with the re-encoding and hierarchical beam search strategies to achieve the objective. (3) We automatically create a dataset named CMWPA for solving complicated MWPs that require both the basic binary operators and the advanced operators. It has been shown that GeDe not only effectively adapts advanced operators but also performs better on three existing MWP

datasets when more operations are involved.

## 2 Related Work

### 2.1 Math Word Problem

Early efforts to solve MWPs use rule-based approaches, which are only able to address a limited number of MWP scenarios (Kushman et al., 2014; Liguda and Pfeiffer, 2012; Roy and Roth, 2018). Deep learning models, on the other hand, are better capable of addressing a wider range of MWPs. The first seq2seq model for MWPs is proposed by Wang et al. (2017). This model employs RNN to encode the problem and produce mathematical expressions. To enhance the seq2seq model, additional techniques have been developed, including reinforcement learning (Huang et al., 2018), template-based methods (Wang et al., 2019), and group attention mechanisms (Li et al., 2019). Seq2tree, a tree structure decoder, is developed by Xie and Sun (2019). It replaces the original sequence decoder and greatly outperforms seq2seq models in terms of performance. KA-S2T (Wu et al., 2020) and MWP-BERT (Liang et al., 2022) inject commonsense knowledge and quantities' properties to improve model performance. In order to encode the relationships between quantities in MWPs, Graph2tree (Li et al., 2020; Zhang et al., 2020) encodes the input problem using graph neural networks.

In addition to the generation models with seq2seq, seq2tree, or graph2tree structures, other efforts use deductive methods to solve MWPs step by step rather than directly generating the entire expression. Cao et al. (2021) represent the calculation process by bDAG and extract the bDAG structure by aggregating quantities and sub-expressions iteratively. Jie et al. (2022) view the task as a complex relation extraction problem and predict the relation of two quantities gradually. Compared with generation methods, deductive methods can easily employ the intermediate values to avoid repetitive generation. We expand the deductive methods to handle more complex advanced operators.

### 2.2 Large-scale Pre-trained Language Model

In-context few-shot learning or even zero-shot learning based on large-scale pre-trained language models, such as GPT-3 (Brown et al., 2020), PaLM (Chowdhery et al., 2022), and OPT (Zhang et al., 2022), has been thoroughly studied for multiple tasks, including math word problem solving (Cobbe et al., 2021; Wang et al., 2022; Wei et al., 2022). This tuning-free methods have achieved promising performance, and their success mainly relies on the reasoning power of large-scale PLMs. However, the reasoning power is extremely expensive due to the large number of parameters, massive pre-training data, carefully designed pre-training objectives, and huge overhead of computational resources. In contrast, we investigate fine-tuning the small models.

## 3 Problem Definition

The goal of MWP is to generate a specific logic form that can be executed to answer the problem $P = \{p_1, p_2, .., p_n\}$ which consists of $n$ tokens and $m$ quantity tokens $Q = \{q_1, q_2, ..., q_m\}$. Some commonsense constants, such as $\pi$ and $e$, may not explicitly appear in the problem; thus, we additionally add them to the quantity set $Q$.

In this paper, we define the multivariate[1] directed acyclic graph (mDAG) as our target logic form, which describes the process of solving MWPs. The nodes of mDAG denote operations that consist of an operator and multiple quantities, and the edges represent the dependency between nodes. Our goal is to generate a operation sequence $O = (o^1, o^2, ..., o^{|O|})$ which can be obtained from the topological ordering of mDAG. $|O|$ is the number of operations. The $t$-th operation is a sequence of tokens $o^t = (a_1^t, a_2^t, ..., a_{|o^t|}^t)$ with each token representing an operator or a quantity. Each operator is selected from the operator set $V$, which is predefined by the provided dataset. Each quantity is choosen from $Q$, which is initialized with the $m$ quantity tokens in $P$ and can gradually grow as the steps of reasoning progress. $|o^t|$ is the number of tokens of the $t$-th operation.

## 4 Approach

### 4.1 Overview

In general, the proposed GeDe method consists of two main components: the re-encoder and decoder. The former aims to jointly encode the problem and quantities, which can support the reuse of intermediate quantities. The latter is designed to generate an operation according to the output of the re-encoder. Since our target is an operation sequence, we need to perform multiple reasoning steps, with each step generating an operation. We illustrate the reasoning process in Figure 2. At each

---

[1] The term "multivariate" means that the operator can receive multiple quantities and output multiple quantities.

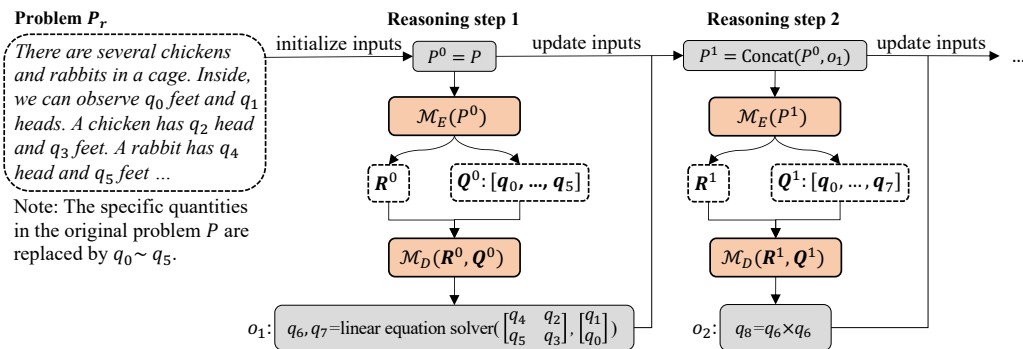

Figure 2: Illustration of iteratively generating the operation sequence by the proposed GeDe. At each reasoning step, GeDe re-encodes the input by adding new intermediate quantities and then generates a new operation.

reasoning step, we update the input sequence by adding new intermediate quantities generated in the previous step. The updated input sequence is fed into the re-encoder and the decoder to generate an operation. The generation process is equipped with a hierarchical beam search strategy to enable both token-level beam search within an operation and operation-level beam search in the whole operation sequence.

## 4.2 Re-Encoder

This section delves into the re-encoder by explaining the input and the encoder respectively.

Since we are only interested in the semantics of the quantities rather than their precise values, we first substitute each quantity in the original problem $P$ with a general special token, $[\text{QTT}_i]$. This leaves $P_r$ devoid of any specific quantities. In order to obtain the encoder's input sequence, $P_{in}^t$, we concatenate $P_r$ with all intermediate quantities, where each quantity signifies its corresponding operation.

We take the example in Figure 2 to explain the input. The given math problem contains six quantities, which are replaced by $[\text{QTT}_0]$ to $[\text{QTT}_5]$. At reasoning step $t$, we have already generated the following operation:

$$[\text{LES}] \begin{bmatrix} [\text{QTT}_2] & [\text{QTT}_4] \\ [\text{QTT}_3] & [\text{QTT}_5] \end{bmatrix} \begin{bmatrix} [\text{QTT}_0] \\ [\text{QTT}_1] \end{bmatrix} \quad (1)$$
$$= [\text{LES}][\text{QTT}_2][\text{QTT}_4][\text{QTT}_3][\text{QTT}_5][\text{QTT}_0][\text{QTT}_1]$$

where LES stands for a multivariant operator of linear equation solver given the operands of a matrix made up of $[\text{QTT}_2]$, $[\text{QTT}_3]$, $[\text{QTT}_4]$, $[\text{QTT}_5]$ and a vector made up of $[\text{QTT}_0]$ and $[\text{QTT}_1]$. In practice, the operation is represented by a sequence that expands the matrix and vector by row. Then we denote the outputs of this operation by two new

quantities $[\text{QTT}_6]$ and $[\text{QTT}_7]$ and concatenate the sequence

$$[\text{QTT}_6][\text{QTT}_7][=][\text{LES}][\text{QTT}_2][\text{QTT}_4]\cdots[\text{QTT}_1] \quad (2)$$

with the original input $P_r$ to obtain $P_{in}^t$.

We instantiate the re-encoder $\mathcal{M}_E$ by a PLM (*e.g.*, BERT or GPT) to represent the input sequence and obtain the reasoning state, *i.e.*,

$$\mathbf{R}^t = \mathcal{M}_E(P_{in}^t), \quad (3)$$

where $\mathbf{R}^t \in R^{N \times H}$ represents the reasoning state at step $t$. $N$ denotes the length of the input sequence and $H$ denotes the hidden size.

For the subsequent generation module, we extract the representation of each quantity from $\mathbf{R}^t$ according to their positions in $P_{in}^t$:

$$\mathbf{Q}^t = \{\mathbf{R}^t[i] \mid i \in I_q\}, \quad (4)$$

where $\mathbf{Q}^t \in R^{M \times H}$, $M$ denotes the number of quantities, $I_q$ saves the indexes of all the quantities in $P_{in}^t$, and $\mathbf{R}^t[i]$ denotes the $i$-th row of $\mathbf{R}^t$.

In summary, the original input is re-encoded with the previously generated intermediate quantities at each reasoning step to update the reasoning state and record all intermediate quantities, which may be reused in the subsequent generation process.

## 4.3 Decoder

We adopt a Gated Recurrent Unit (GRU) network (Chung et al., 2014) combined with the attention mechanism (Vaswani et al., 2017) as the decoder $\mathcal{M}_D$. Following the majority of the earlier works (Liang et al., 2022; Tan et al., 2021; Xie and Sun, 2019), we choose GRU instead of transformer for a fair comparison. Although some works choose pretrained transformer (Shen et al., 2021), their performance might not be improved due to the larger parameters but limited labeled data.

**Operation Generation.** The decoder aims to provide an operation $o^t = (a_1^t, a_2^t, ..., a_{|o^t|}^t)$ at each reasoning step $t$. To enable the auto-regressive generation, we insert a special beginning token ([BOS]) before the first token $a_1^t$ and add a special ending token ([EOS] or [EOO]) after the last token $a_{|o^t|}^t$ to re-create $o^t = (a_0^t, a_1^t, a_2^t, ..., a_{|o^t|+1}^t)$. While [EOS] only signifies the termination of the current operation, [EOO] stands for the final token of the complete operation sequence. The hidden state $\mathbf{h}_i^t$ of each token $a_i^t$ can be obtained by $\mathbf{h}_i^t = \text{GRU}(\mathbf{h}_{i-1}^t, \mathbf{a}_i^t)$ where $\mathbf{h}_{i-1}^t \in R^{1 \times H}$ represents the hidden state of the previous step, $\mathbf{h}_0^t$ is initialized from the hidden state of the [CLS] token produced by the encoder, and $\mathbf{a}_i^t \in R^{1 \times H}$ is the representation of the token $a_i^t$. Next, using $\mathbf{h}_i^t$ as the query to attend to current reasoning state $\mathbf{R}^t$, we obtain the attention-enhanced state $\mathbf{A}_i^t = \text{MHA}(\mathbf{h}_i^t, \mathbf{R}^t)$, where MHA denotes multi-head attention (Vaswani et al., 2017). Finally, we determine the likelihood of the output token by determining how well $\mathbf{A}_i^t$ resembles the representation of quantities and operators, *i.e.*,

$$p(a_i^t|o^{<t}, a_{<i}^t, P) = \text{softmax}(\mathbf{A}_i^t([\mathbf{V} \mid \mathbf{Q}^t])^T), \quad (5)$$

where $o^{<t}$ represents $o^1, o^2, ..., o^{t-1}$ before reasoning step $t$, $a_{<i}^t$ represents $a_0^t, a_1^t, \cdots, a_{i-1}^t$ before the $i$-th token of step $t$, $\mid$ is the matrix concatenation operator, $\mathbf{V} \in R^{|V| \times H}$ and $\mathbf{Q}^t \in R^{M \times H}$ denote the representations of operators and $t$-th step's quantities respectively. When obtaining a new operation $o^t$, we can determine the number of new quantities by the operator in $o^t$ and record these new intermediate quantities for the subsequent reasoning steps. When [EOS] has the highest probability, the decoding process of the current operation ends but a new operation generation starts instead. When [EOO] has the highest probability, the entire decoding process is complete.

**Training Objective.** Given a problem $P$ and its ground truth operation sequence $O$, we maximize the probability of generating $O$ by $P$, *i.e.*,

$$p(O|P) = \prod_{t=1}^{|O|} \prod_{i=1}^{|o^t|+1} p(a_i^t|o^{<t}, a_{<i}^t, P). \quad (6)$$

### 4.4 Hierarchical Beam Search

To enhance the generation quality during inference, beam search is used in many generation tasks as a refined version of greedy search (Tillmann and Ney, 2003). However, using beam search in the deductive methods is difficult because the search space of the operation sequence is nested. In other words, we need to generate each operation based on tokens and generate the entire operation sequence based on operations. Therefore, previous deductive methods (Cao et al., 2021; Jie et al., 2022) only adopt the greedy search and leave the implementation of the beam search as further work. To address this challenge, we propose a *hierarchical beam search* strategy. Compared with the traditional beam search, the hierarchical beam search can control the generation process at two levels.

Specifically, the hierarchical beam search consists an inner beam search and an outer beam search. The former is a standard beam search which seeks a series of tokens to form a candidate operation. The latter is designed to search a complete operation sequence. The beam score of the inner beam search purely relies on the probabilities of tokens predicted by the decoder. Suppose the $t$-th step generates $l$ tokens, the inner beam score $ibs^t$ is calculated as:

$$ibs^t = \log \prod_{i=1}^{l} p(a_i^t)^{\frac{1}{l}} = \frac{1}{l} \sum_{i=1}^{l} \log p(a_i^t), \quad (7)$$

where $p(a_i^t)$ is computed by Eq (5). We use the inner beam scores of generated operations to approximate the distribution of operations to support the outer beam search. The probability of the $t$-th operation $o^t$ can be calculated as the softmax score of its inner beam score, *i.e.*,

$$p(o^t) = \text{softmax}(\exp(ibs^t)). \quad (8)$$

Suppose the entire operation sequence contains $T$ operations, the outer beam score is computed as:

$$obs = \log(\prod_{t=1}^{T} p(o^t))^{\frac{1}{T}} = \frac{1}{T} \sum_{t=1}^{T} \log p(o^t). \quad (9)$$

Algorithm 1 presents the hierarchical beam search algorithm. Each outer beam is denoted by the symbol $beam$, which keeps track of both the current operation sequence and the beam score. The empty operation sequence and score of zero are used to construct the initial outer beam initially (line 1). Then, we iteratively expand outer beams until they are all finished, *i.e.*, all the outer beams are terminated with [EOO] (line 4-14). For

**Algorithm 1** Hierarchical Beam Search

---
**Input:** Math World Problem $P$, Beam size $K$
**Output:** $beams$ with Top-K operation sequences
1: $beams \leftarrow [InitialBeam]$;
2: **while** not all beams are over **do**
3:    $beams_n \leftarrow []$;
4:    **for** $beam$ in $beams$ **do**
5:       **if** $beam$ is over **then**
6:          $beams_n.append(beam)$;
7:       **else**
8:          $ops \leftarrow \text{InnerBeamSearch}(P, beam, K)$;
9:          **for** $op$ in $ops$ **do**
10:             $beam_{new} \leftarrow \text{Extend}(beam, op)$;
11:             $beams_n.append(beam_{new})$;
12:          **end for**
13:       **end if**
14:    **end for**
15:    $beams \leftarrow \text{GetTopK}(beams_n, K)$;
16: **end while**

---

each extensible outer beam, we search candidate operations $ops$ using the inner beam search (line 8). The inner and the outer beam search share the same beam size $K$. Next, we extend outer beams with these candidate operations (line 9-12). At the end of each step, we only maintain the top-K outer beams according to their scores computed by Eq. (9) (line 15). Finally, $beams$ save the top-K operation sequences. We discuss the complexity of GeDe in Appendix A.1

### 4.5 Decoding Constraint

Logic forms need to obey clear grammatical rules. In order to guarantee the validity of the output, we provide two constraint strategies, one during and one after the decoding process. Inspired by PICARD (Scholak et al., 2021), an incremental grammar checker proposed for Text-to-SQL task, the constraint strategy during the decoding process is to filter out illegal beams at each decoding step in the inner beam search to prevent potential syntax errors in the generated operation. For example, when we detect that the current token generation step needs to generate an operator, we will reject all non-operators. Following (Jie et al., 2022), the after decoding constraint strategy eliminates candidate operations that are improbable to exist in real-world mathematical problems, such as "$[\text{QTT}_i] - [\text{QTT}_i]$" and "$[\text{QTT}_i]^{[\text{QTT}_i]}$".

## 5 Experiments

In this section, we establish a dataset for multivariant advanced operators and show that the proposed GeDe is capable of doing these types of operations successfully. We also conduct experiments on four widely-adopted MWP datasets to show the effectiveness of our model on binary operations.

### 5.1 Experimental Setup

**Datasets.** We consider four MWP datasets including our created CMWPA and three widely-used existing MWP datasets: MAWPS (Koncel-Kedziorski et al., 2016), Math23k (Wang et al., 2017), MathQA (Amini et al., 2019), and SVAMP (Patel et al., 2021). We use CMWPA to verify the validity of multivariate operations. Following (Tan et al., 2021), we perform pre-processing to filter out unsolvable problems. In all the datasets, we take into account the basic binary operators addition ($+$), subtraction ($-$), multiplication ($\times$), division ($\div$), and exponentiation ($\char`\^$). For advanced operators used in the CMWPA dataset, we consider the linear equation solver, the quadratic function extremum solver, and the quadratic function integral solver. Appendix A.2 presents the statistics for each dataset.

**Evaluation Metric.** Following previous work (Jie et al., 2022), we compare the predicted and the gold answer to calculate the accuracy as the evaluation metric. We parse out the operator and operands from the model predicted expression sequence and then use the corresponding operator executor to calculate the answers. We explain the details of the parsing and execution in Appendix A.3.

**Implementation Details.** We adopt RoBERTa-base[2] (Liu et al., 2019) as our re-encoder for English datasets, and Chinese-RoBERTa-base[3] (Cui et al., 2020) for Chinese datasets. The purpose of using the Roberta model is to make a more fair comparison with previous work. We can also use unidirectional attention models (*e.g.*, GPT). We use AdmaW to optimize the loss function with a learning rate of 2e-5, a weight decay of 1e-2, and a batch size of 8. During inference, the beam size $K$ is set to 4 by default. For CMWPA, Math23K, MathQA, and SVAMP we report accuracy on their test set. For MAWPS and Math23k, we follow previous works and also report 5-fold cross-validation performance. We conduct all experiments with a RTX 3090 (24G) GPU.

---

[2]https://huggingface.co/roberta-base
[3]https://huggingface.co/hfl/chinese-roberta-wwm-ext

| | Model | MAWPS 5-fold | Math23k Test Set | Math23k 5-fold | MathQA Test Set | SVAMP Test Set |
|---|---|---|---|---|---|---|
| S2S | GroupAttn (Li et al., 2019) | 76.1 | 69.5 | 66.9 | - | 21.5 |
| | mBERT+LSTM (Tan et al., 2021) | - | 75.1 | - | 77.1 | - |
| | RoBERTaGen (Lan et al., 2022) | 88.4 | - | 76.9 | 76.6 | 30.3 |
| | Generate&Rank (Shen et al., 2021) | 84.0 | **85.4** | **84.3** | - | - |
| | GTS (Xie and Sun, 2019) | 82.6 | 75.6 | 74.3 | - | 41.0 |
| S2T/G2T | Graph2Tree (Zhang et al., 2020) | 85.6 | 77.4 | 75.5 | 69.5 | 43.8 |
| | HMS (Lin et al., 2021) | 80.3 | 76.1 | - | - | - |
| | MultiE&D (Shen and Jin, 2020) | - | 78.4 | 76.9 | - | - |
| | BERT-CL (Li et al., 2022) | - | 82.4 | - | 73.8 | - |
| | MWP-RoBERTa (Liang et al., 2022) | - | 84.5 | 82.0 | 76.6 | - |
| DR | RoBERTa-DR (Jie et al., 2022) | 92.0 | 85.1 | 83.0 | 78.6 | **47.3** |
| | GeDe | **92.3** | **85.4** | 84.2 | **81.5** | 45.7 |

Table 1: Accuracy on three existing MWP datasets (%).

## 5.2 Experiment on CMWPA

The existing MWP datasets only use basic binary operators as target logic form. Rewriting these logic forms to support advanced operators is expensive. Therefore, based on handcraft templates, we create a synthetic dataset named **CMWPA** (Complex Math Word Problems with Advanced operators).

To create the CMWPA dataset, we first define needed operators which include five binary operators (addition ($+$), subtraction ($-$), multiplication ($\times$), division ($\div$), and exponentiation (ˆ)), as well as three advanced operators, which can be used to solve linear equations (the [linear equation solver] operator), find the maximum value of quadratic functions (the [quadratic function extremum solver] operator), and find the definite integrals of quadratic functions (the [quadratic function integral solver] operator). For each operator, we write one or more templates to generate a text description and its operation. We only consider the quadratic function because the operations related to the quadratic function can be transformed to a series of binary operations for training the baseline model. The templates of **CMWPA** is described in Appendix A.4. In this dataset, for each problem, we provide two types of logic forms: multivariate operation sequence and binary operation sequence. An example is given in Appendix Table 5.

We conduct experiments on CMWPA to demonstrate that using advanced operators to solve complex MWPs is more effective than only using basic binary operators. Concretely, our proposed GeDe is applied to generate multivariate operation sequences. Then for fair comparison, we adopt GeDe to generate binary operation sequence.

**Experiment Results.** Table 2 shows the accuracy and inference time on CMWPA, using mDAG as

| Logic Form | Accuracy | Inference Time |
|---|---|---|
| BET | 32.0 | 600 ms/per sample |
| mDAG | 95.0 | 400 ms/per sample |

Table 2: Accuracy (%) and time cost on CMWPA of GeDe with different annotation (BET: Binary Expression Tree, mDAG: Multivariate Directed Acyclic Graph).

annotation, GeDe achieves 95.0% accuracy, which indicates that our proposed method can effectively support advanced operators to generate the multivariate operation sequence. However, when using the binary expression tree as the generation target, GeDe only achieves 32.0% accuracy. Because the average number of advanced operators in multivariate operation sequences is 2.98, which is significantly less than the average number of binary operators (*i.e.*, 35.03) in binary expressions tree, using advanced operators to solve MWPs can essentially reduce the learning difficulty and lead to improved both the accuracy and efficiency.

## 5.3 Experiment on Existing MWP Datasets

**Baselines.** The baselines can be broadly categorized into four groups, sequence-to-sequence(S2S), sequence-to-tree(S2T), graph-to-tree(G2T), and deductive-reasoning(DR), where the first three of these are all generation-based methods but are instantiated with different encoders or decoders. We select baselines having reported the performances on at least one of the three datasets.

**Experiment Results.** We start by running tests on MAWPS and Math23k. As shown in Table 1, our model achieves promising performance on both the datasets compared to previous state-of-the-art (SOTA) methods. Given that MAWPS only has an average of 1.41 binary operations, the proposed

GeDe only slightly improves 0.3% accuracy on MAWPS compared to earlier baselines. This is not enough to demonstrate the benefits of the proposed model. On Math23k, GeDe performs equally well as the earlier SOTA method Generate&Rank. However, Generate&Rank fine-tunes a mBART-large (Liu et al., 2020) model with 610M parameters. In contrast, GeDe only involves 126M parameters and thus reveals a better parameter-efficiency. We further evaluate our method on MathQA, the most challenging MWP dataset with an average of 4.25 binary operations, and show results in Table 1. Our model greatly beats all baselines (+2.9%), which demonstrates the model's efficacy in handing complex MWPs. In summary, on three existing MWP datasets, the performances of GeDe are on par or better than those of the closest competitors. SVAMP is also a challenging dataset that is manually created to evaluate a model's robustness. On this dataset, GeDe achieves an accuracy of 45.7%, which can outperform the vast majority of baselines except the DR model.

In addition, we conduct experiments based on Roberta-large on the Math23k dataset. The model achieves an accuracy of 86.7% on the Math23K test set. Using Roberta-large improves the accuracy by 1.3% over using Roberta-base. This shows that using a larger PLM improves the performance of our method and outperforms the baseline Generate & Rank model on the Math23K test set.

To further highlight the advantages of the proposed GeDe, following (Jie et al., 2022), we provide a fine-grained analysis on MathQA based on various numbers of operations. To be more specific, we compare our model with the most powerful baseline RoBERTa-DR (Jie et al., 2022) and display the analysis results in Table 3. We observe that GeDe performs better on samples with 1, 3, and 4 operations, particularly on samples with at least 5 operations. This comparison indicates our model is more robust to problems requiring more reasoning steps, because the designed re-encoder can capture adequate interactions between the newly produced quantities and the original problem.

## 5.4 Ablation Study

In this section, we take a thorough ablation study on MathQA dataset to verify the effectiveness of the re-encode and the hierarchical beam search strategies in the proposed GeDe.

**Effect of Re-encoder.** The proposed re-encoder

| # Operations | RoBERTa-DR | GeDe |
|---|---|---|
| 1 | 77.4 | **78.0** |
| 2 | **83.5** | 81.8 |
| 3 | 83.4 | **85.1** |
| 4 | 81.7 | **84.0** |
| ≥5 | 71.4 | **77.5** |
| Overall | 78.6 | **81.5** |

Table 3: Fine-grained accuracy on MathQA (%).

| Model variant | Accuracy |
|---|---|
| GeDe | **81.5** |
| - w/o dynamic quantity embeddings | 80.3 |
| - w/o re-encoder | 75.8 |
| - w/o hierarchical beam search | 81.0 |

Table 4: Ablation study on MathQA (%).

in Section 4.2 can update both new quantities and reasoning state at each reasoning step. We investigate the two functions respectively.

Instead of using dynamic quantity embeddings, we develop a variant model with static quantity embeddings. In other words, instead of having distinct embeddings updated based on various contexts in various math problems, [QTT$_i$] in various math problems is assigned a unified embedding that is updated globally. Note we keep re-encoding the original problem with the newly produced quantities at each step $t$, but only the updated reasoning state $\mathbf{R}^t$ is leveraged. The comparison results in Table 4 show that without the dynamic quantity embeddings, the performance drops 1.2% on MathQA's test set. Since different MWPs' quantities reflect different semantics, it is preferable for them to be dynamically updated with their contexts.

Then we completely remove the re-encoder and only allow the encoder to encode the original problem. Instead, we directly use the hidden state in the decoder's GRU network to represent the reasoning state. Table 4 shows that without the re-encoder, the performance drops 5.7%. In this variant model, although the quantities are dynamically updated according to various problems, the interactions between the quantities and the input problem are not fully exploited as the re-encoder does.

**Effect of Hierarchical Beam Search.** Previous deductive methods (Cao et al., 2021; Jie et al., 2022) generate the operation sequence based on hierarchical greedy search, and regard the implementation of beam search as a future challenge. We imple-

ment hierarchical beam search in our GeDe to improve greedy search. We compare them, where the beam size is set to 1 to create a greedy search. As shown in Table 4, when the hierarchical beam search is disabled (beam size = 4) and replaced with the hierarchical greedy search (beam size = 1), the performance drops $0.5\%$. By observing the inner and outer beam scores in the generation process, for most of the samples, we find that the score of the first beam is significantly greater than that of the remaining beams, resulting in a relatively small gap between greedy and beam search. This problem, also referred to as neural networks' "over-confidence", has been studied by some works (Miao et al., 2021; Wang et al., 2021). Such improvement is left in the further.

## 6 Conclusion and Future Work

This paper proposes a multivariant direct acyclic graph (mDAG) to describe a math expression in order to handle the advanced multivariant operators. Then to generate the topological ordering of mDAG, we propose a generation model equipped with a re-encode strategy to keep the deductive property but avoid the expensive enumeration of existing deductive methods. A hierarchical beam search algorithm is implemented to enable the inner token and outer operation searches. Extensive experiments on three standard datasets and one automatically created dataset demonstrate the proposed model's advantage in solving math problems with binary operators and advanced operators.

## 7 Limitations

From the time complexity analysis in Appendix A.1, we can see that our model will face the efficiency issue when it needs to generate a long operation sequence. At the same time, the re-encode module needs to concatenate the problem description with generated operations, which may reach the input length limit of PLM. Therefore, our future work will study how to compress the input sequence during the generation process to address above issues.

## 8 Ethics Statement

For many years, public opinion has debated the pros and cons associated with artificial intelligence technology. One consensus is that advances in technology may be used in a variety of scenarios, leading to different influences. To provide an ethical analysis of this work and others on the same line, we will address three aspects: the possible positive or negative effects of our work, the impact of harmful information in datasets, and the equality and differences between different languages.

First, the point of studying MWP is to explore the mathematical reasoning capabilities of artificial intelligence (Wei et al., 2022). However, the developed models may still be applied to harmful aspects, such as cheating in math exams.

On the other hand, the presence of harmful information in the training data may lead the model to learn some implicit biases (Liang et al., 2021; Steed et al., 2022). In our experiments, for the three existing datasets, we exactly follow the experimental setup of previous works to pre-process and remove the potential harmful information. For our manually created dataset CMWPA, our templates also do not contain any harmful information. However, in the inference phase, our model cannot reject answers when the user provides malicious input. Therefore, we need to employ extra efforts to avoid this issue when the model is deployed online.

Finally, we use both English and Chinese datasets in our experiments to respect linguistic equality and better take into account language differences. The experimental results validate the robustness of our model across languages. Nevertheless, English and Chinese are the two most popular languages, and we should make greater efforts to concentrate on and preserve the development of minor languages in the field of natural language processing (Zhang et al., 2021).

## Acknowledgments

This work is supported by National Natural Science Foundation of China (62322214, 62072460, 62172424,62276270); Beijing Natural Science Foundation (4212022); the Public Computing Cloud at Renmin University of China.

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

# A  Appendix

## A.1  Complexity Analysis

Consider a math problem that has $n$ words in the problem description and $|O|$ operations in the solving process. A total of $\kappa$ words are needed to describe the $|O|$ operations, *i.e.*, $\kappa = \sum_{t=1}^{|O|} |o^t|$. GeDe needs to perform $|O|$ operation re-encode steps and $\kappa$ token generation steps. For the $\tau$-th re-encode step, its computational complexity is $O((n + \sum_{t=1}^{\tau-1} |o^t|)^2)$. For generating the tokens in $o^\tau$, its computational complexity is $O(|o^\tau| * (n + \sum_{t=1}^{\tau-1} |o^t|))$. Therefore, the overall time complexity is $\sum_{\tau=1}^{|O|}(n + \sum_{t=1}^{\tau-1} |o^t|)^2) + |o^\tau| * (n + \sum_{t=1}^{\tau-1} |o^t|) < O(|O| * n^2 + \kappa * (n + \kappa))$. If we use the unidirectional attention model as re-encoder, the complexity can be lowered to $O(n^2 + \kappa * (n + \kappa))$, which is the same as what the current seq2seq generation methods achieve. The additional time complexity is acceptable because $|O|$ is typically not very large.

## A.2  Datasets Statistics

The statistics of datasets are presented in Table 6. CMWPA is a synthetic English dataset with 1000 training samples and 100 test and validation samples. MAWPS and MathQA are public English MWP datasets that contain 1.9K math problems and 20K math problems, respectively. Math23K is a public Chinese MWP dataset that contains 23K math problems. We use the average number of operations to assess the difficulty of a MWP dataset. As we can see, MAWPS is the simplest dataset because almost all problems require only one or two operations. MathQA is the most challenging dataset, requiring more operations and, hence, more steps in the reasoning process to obtain the answer. SVAMP is also a challenging dataset that is manually created to evaluate a model's robustness. They apply variations to the instances sampled from MAWPS. Such variations could include adding extra quantities, swapping the positions between noun phrases, etc.

## A.3  Parsing and Execution

Due to the existence of higher-order operators, the way we calculate the answer is different from previous works. We implement the corresponding solving function using Python for each pre-defined operator, which is also included in our published code. During inference, for the generated operation sequence, we sequentially calculate the returned quantities for each operation. Naturally, the returned quantities of the last operation denote the answer to the problem. For a generated operation, we first parse out its operator and several operands. Then, we call the solving function corresponding to the operator to obtain the returned quantities.

## A.4  CMWPA Templates

We show the templates corresponding to the advanced operators as follows.

Two templates for the [linear equation solver] operator:

- Text description: [q_0] * [o_0] + [q_1] * [o_1] = [q_4]; [q_2] * [o_0] + [q_3] * [o_1] = [q_5].

  Operation: [linear equation solver] [q_0] [q_1] [q_2] [q_3] [q_4] [q_5]

- Text description: Determine [o_0], [o_1] as the result of inverse of matrix [ [ [q_0] , [q_1] ] , [ [q_2] , [q_3] ] ] times vector [ [q_4] , [q_5] ].

Operation: [linear equation solver] $[q_0]$ $[q_1]$ $[q_2]$ $[q_3]$ $[q_4]$ $[q_5]$

One template for the [quadratic function integral solver] operator:

- Text description: Determine $[o_0]$ as the definite integral of quadratic function $[q_0] * x\hat{\ }2 + [q_1] * x + [q_2]$ between the intervals $[q_3]$ and $[q_4]$.

  Operation: [quadratic function integral solver] $[q_3]$ $[q_4]$ $[q_0]$ $[q_1]$ $[q_2]$

One template for the [quadratic function extremum solver] operator:

- Text description: Determine $[o_0]$ as the the extremum value of quadratic function $[q_0] * x\hat{\ }2 + [q_1] * x + [q_2]$.

  Operation: [quadratic function extremum solver] $[q_0]$ $[q_1]$ $[q_2]$

Based on the templates, we generate a sample as follows. First, we randomly initialize a candidate set of quantities. Then, we randomly select a template and fill in slots by randomly selecting quantities from the quantity candidate set. We re-input the returned quantities of the operation into the candidate set and repeat the above process several times. In this way, a problem description and its operation sequence are generated. We also convert the operation sequence into a pre-order binary expression as another type of annotation for training the seq2seq baseline.

### A.5 CMWPA Example

We provide a sample of CMWPA in Table 5. This sample is initialized with 6 quantities and involves four types of operators: the subtraction operator, the [linear equation solver] operator, the [quadratic function integral solver] operator, and the [quadratic function extremum solver] operator. Two types of annotations are provided: the multivariant operation sequence and the pre-order binary expression (pre-order binary expression can be transformed into a binary operation sequence (bDAG) or a binary expression tree). For each operation in the multivariant operation sequence, we provide the operation, the input quantities, and the returned output quantities.

**Problem**:
Given $[q_0] = 0.23$ . $[q_1] = 0.43$ . $[q_2] = 0.18$ . $[q_3] = 0.26$ . $[q_4] = 0.71$ . $[q_5] = 0.85$ . Determine $[q_6]$ as the $[q_4]$ minus $[q_5]$ . Determine $[q_7]$ $[q_8]$ as the result of inverse of matrix $[ [ [q_4] , [q_3] ] , [ [q_2] , [q_5] ] ]$ times vector $[ [q_0] , [q_6] ]$ . Determine $[q_9]$ as the definite integral of quadratic function $[q_6] * x^2 + [q_7] * x + [q_5]$ between the intervals $[q_1]$ and $[q_8]$ . Determine $[q_{10}]$ as the the extremum value of quadratic function $[q_8] * x^2 + [q_6] * x + [q_9]$ . Output the value of $[q_{10}]$ .

**Multivariant Operation Sequence**:

1. operation$_1$: [-, [QTT$_4$], [QTT$_5$]]

   returned quantities of operation$_1$: [[QTT$_6$]]

2. operation$_2$: [[linear equation solver], [QTT$_4$], [QTT$_3$] , [QTT$_2$] , [QTT$_5$] , [QTT$_0$] , [QTT$_6$]]

   returned quantities of operation$_2$: [[QTT$_7$], [QTT$_8$]]

3. operation$_3$: [[quadratic function integral solver] , [QTT$_1$] , [QTT$_8$] , [QTT$_6$] , [QTT$_7$] , [QTT$_5$]]

   returned quantities of operation$_3$: [[QTT$_9$]]

4. operation$_4$: [[quadratic function extremum solver] , [QTT$_8$] , [QTT$_6$] , [QTT$_9$]]

   returned quantities of operation$_4$: [[QTT$_{10}$]]

**Pre-order binary expression**:
+ , * , / , - , * , [QTT$_4$] , [QTT$_0$] , * , [QTT$_2$] , - , [QTT$_4$] , [QTT$_5$] , - , * , [QTT$_4$] , [QTT$_5$] , * , [QTT$_2$] , [QTT$_3$] , ^ , * , [c3] , / , - , [QTT$_4$] , [QTT$_5$] , * , [c1] , / , - , * , [QTT$_4$] , [QTT$_0$] , * , [QTT$_2$] , - , [QTT$_4$] , [QTT$_5$] , - , * , [QTT$_4$] , [QTT$_5$] , * , [QTT$_2$] , [QTT$_3$] , [c1] , + , * , - , [QTT$_4$] , [QTT$_5$] , * , [c3] , / , - , [QTT$_4$] , [QTT$_5$] , * , [c1] , / , - , * , [QTT$_4$] , [QTT$_0$] , * , [QTT$_2$] , - , [QTT$_4$] , [QTT$_5$] , - , * , [QTT$_4$] , [QTT$_5$] , * , [QTT$_2$] , [QTT$_3$] , - , - , + , * , / , - , [QTT$_4$] , [QTT$_5$] , [c2] , ^ , / , - , * , [QTT$_4$] , [QTT$_0$] , * , [QTT$_2$] , - , [QTT$_4$] , [QTT$_5$] , - , * , [QTT$_4$] , [QTT$_5$] , * , [QTT$_2$] , [QTT$_3$] , [c2] , + , * , / , / , - , * , [QTT$_5$] , [QTT$_0$] , * , [QTT$_3$] , - , [QTT$_4$] , [QTT$_5$] , - , * , [QTT$_4$] , [QTT$_5$] , * , [QTT$_2$] , [QTT$_3$] , [c1] , ^ , / , - , * , [QTT$_4$] , [QTT$_0$] , * , [QTT$_2$] , - , [QTT$_4$] , [QTT$_5$] , - , * , [QTT$_4$] , [QTT$_5$] , * , [QTT$_2$] , [QTT$_3$] , [c1] , * , [QTT$_5$] , / , - , * , [QTT$_4$] , [QTT$_0$] , * , [QTT$_2$] , - , [QTT$_4$] , [QTT$_5$] , - , * , [QTT$_4$] , [QTT$_5$] , * , [QTT$_2$] , [QTT$_3$] , + , * , / , - , [QTT$_4$] , [QTT$_5$] , [c2] , ^ , [QTT$_1$] , [c2] , + , * , / , / , - , * , [QTT$_5$] , [QTT$_0$] , * , [QTT$_3$] , - , [QTT$_4$] , [QTT$_5$] , - , * , [QTT$_4$] , [QTT$_5$] , * , [QTT$_2$] , [QTT$_3$] , [c1] , ^ , [QTT$_1$] , [c1] , * , [QTT$_5$] , [QTT$_1$]

Table 5: A sample of CMWPA. [QTT$_i$] represents the $i$-th quantity, [c1], [c2], and [c3] represent three constants 1, 2, and 3 respectively.

| Dataset | #Train | #Dev | #Test | Avg. #Operations | Avg. PDL | Operation Types | Language |
|---------|--------|------|-------|------------------|----------|-----------------|----------|
| CMWPA | 1,000 | 100 | 100 | 2.98 | 329.55 | Basic & Advanced | English |
| MAWPS | 1,589 | 199 | 199 | 1.41 | 299.31 | Basic | English |
| Math23k | 21,162 | 1,000 | 1,000 | 2.27 | 156.28 | Basic | Chinese |
| MathQA | 16,191 | 2,411 | 1,605 | 4.25 | 374.89 | Basic | English |
| SVAMP | 3,138 | - | 1,000 | 1.3 | 159.6 | Basic | English |

Table 6: Detailed statistics of all datasets. PDL means problem description length.