# OpenReview forum: "A Generation-based Deductive Method for Math Word Problems"
_EMNLP/2023/Conference — EMNLP 2023 Main_

### Official Review · Reviewer_asVd · 2023-08-03

**Soundness:** 3

**Excitement:**

4: Strong: This paper deepens the understanding of some phenomenon or lowers the barriers to an existing research direction.

**Missing References:**

Measuring Mathematical Problem Solving With the MATH Dataset, Hendrycks et al., NeurIPS 2021

A Diverse Corpus for Evaluating and Developing English Math Word Problem Solvers, Miao et al. ACL-2020


**Paper Topic And Main Contributions:**

This paper presents a generation-based deductive (GeDe) approach to construct a multivariate directed acyclic graph (mDAG) for MWP solving to address the repeated subtree generation issue and the binary tree expression limitation. The GeDe constructs a mDAG by generating an operation sequence iteratively. GeDe re-encodes the input for each reasoning step by adding new intermediate quantities (concatenating the original MWP) and decodes to generate a new multivariate operation. A hierarchical beam search (operation candidates seeking and operation sequence selecting) is also proposed to improve the model performance. Experimental results demonstrate the effectiveness of three public datasets and one newly released CMWPA dataset.

**Questions For The Authors:**

Q1: Lines #272-#274: How to obtain the representation of the intermediate quantities (such as [QQT_6] and [QQT_7] if they are new quantities)?

Q2: Table 1: Missing related dataset and experimental results descriptions of SVAMP.

Q3: Line #473: CMWPA is a collection of number-word questions synthesized by artificially handcrafted templates that do not contain natural language descriptions in the real world. There are publicly challenging MWP datasets (such as Math or ASDIV). Did the authors conduct experiments on these datasets?


**Reasons To Accept:**

The proposed GeDe approach (includes dynamic quantity embeddings, re-encoder, and multivariate operation mechanisms) is promising, and the experimental results illustrate the effectiveness of solving challenging MWPs. Insights of related experiments could be beneficial for future research on MWP solving.

**Reasons To Reject:**

To demonstrate the effectiveness of the proposed method, the authors additionally construct a synthetic data set instead of using public datasets with challenging MWPs. Consequently, related issues (such as Q3) are to be discussed and explored. Furthermore, the evaluation results of SVAMP in Table 1 are unclear and required to be explained.

**Reproducibility:**

3: Could reproduce the results with some difficulty. The settings of parameters are underspecified or subjectively determined; the training/evaluation data are not widely available.

**Reviewer Confidence:**

4: Quite sure. I tried to check the important points carefully. It's unlikely, though conceivable, that I missed something that should affect my ratings.

**Typos Grammar Style And Presentation Improvements:**

(1)Table2, Row1: “bDAG” -> BET?

(2)Lines #717-#721. Missing the reference source.

(3)Table 6: SVAMO -> SVAMP

---

> ### Author Rebuttal · Authors · 2023-08-26
>
> Q1: We have detailed the representation of new quantities through equations 2, 3, and 4. In equation 2, new quantities are represented as the output of an operation. Subsequently, as illustrated in equation 3, this expression from equation 2 is concatenated with the original input and then re-encoded to yield $R^t$. Lastly, as depicted in equation 4, we extract the vectors associated with these new quantities within $R^t$ to serve as their representations.
>
> Q2: Thank you for pointing out the issue, and we will follow up by revising our paper.
>
> SVAMP is a more challenging dataset that is manually created to evaluate a model’s robustness. They apply variations to the instances sampled from MAWPS. Such variations could include adding extra quantities, swapping the positions between noun phrases, etc. On this dataset, GeDe can outperform the vast majority of baselines except the DR model.
>
> Q3:  Evaluating our model on these two datasets necessitates the implementation of a multitude of intricate parsers and solvers to process and execute the mathematical operations it generates. While this task falls outside the scope of our research forcus and the workload involved is substantial, we will defer it to future work.

---

### Official Review · Reviewer_SjPM · 2023-08-03

**Soundness:** 4

**Excitement:**

3: Ambivalent: It has merits (e.g., it reports state-of-the-art results, the idea is nice), but there are key weaknesses (e.g., it describes incremental work), and it can significantly benefit from another round of revision. However, I won't object to accepting it if my co-reviewers champion it.

**Paper Topic And Main Contributions:**

The work proposed a multivariate acyclic graph expression and a generation-based model with a re-encoder and decoder to generate the operation sequence  in an iterative manner.

**Questions For The Authors:**

Did you conduct experiments with alternative methods on your proposed CMWPA dataset to ensure a fair comparison of various solving techniques, particularly S2S approaches, given that binary or advanced operations should not affect S2S methods? Additionally, did you test your models with Roberta-large, does it improve the performance? It will give us more details about the extent to which your proposed method can achieve, as well as another fair comparison with previous sota models, e.g. generate&rank.

**Reasons To Accept:**

The writing is clear and they performed an ablation analysis. In addition, they collected a dataset to show the effectiveness of their proposed method.

**Reasons To Reject:**

Although the study includes the use of a re-encoder, hierarchical beam search, and quantity embeddings, the improvements in performance is limited. Furthermore, the authors did not report the setting of random seeds or the average performance during multiple experiments.

**Reproducibility:**

3: Could reproduce the results with some difficulty. The settings of parameters are underspecified or subjectively determined; the training/evaluation data are not widely available.

**Reviewer Confidence:**

3: Pretty sure, but there's a chance I missed something. Although I have a good feel for this area in general, I did not carefully check the paper's details, e.g., the math, experimental design, or novelty.

---

> ### Author Rebuttal · Authors · 2023-08-26
>
> The dataset CMWPA is primarily used to demonstrate that using mDAG as the output logic form is superior to using bDAG as the logic form, rather than to compare the performance of different models. We complement the performance of Generate&Rank on CMWPA. The accuracy of the model is 97% in the case of using mDAG annotations, while it is 32% in the case of using bDAG annotations. This further illustrates the advantages of mDAG.
>
> We replaced Roberta-large with Roberta-base in our model and evaluated it on the Math23K dataset. The model achieved an accuracy of 86.7% on the Math23K test set. Using Roberta-large resulted in a 1.3% improvement in accuracy compared to using Roberta-base. This demonstrates that using Roberta-large in our model  enhances performance and outperforms the baseline Generate & Rank model on this Math23K test set.

---

### Official Review · Reviewer_vqVh · 2023-08-05

**Soundness:** 4

**Excitement:**

4: Strong: This paper deepens the understanding of some phenomenon or lowers the barriers to an existing research direction.

**Missing References:**

None.

**Paper Topic And Main Contributions:**

This paper develops a new model that does deduction on math word problems by generating operations. They proposed a model called GeDe, which generates an operation for each deduction step using mDAG structure. To see the effect of their proposal, they tested their model on the synthesized dataset CWMPA and other well-known public benchmarks. Also, they conducted an ablation study to verify whether the components proposed are working as expected. Through these experiments, they showed that their model can handle the N-ary operators properly (as expected) and achieves the state-of-the-art performance using less parameters.

**Questions For The Authors:**

Question A. You've cited Vaswani's work about multi-head attention in line 300, but the equation 5 seems like a single-head attention. Which thing did you use? If you used single-head attention, then why?

Question B. How did you construct CWMPA dataset? What is the base data to synthesize the dataset? I know that the authors already stated it on the appendix, but I think this should be appear at the main paper.

**Reasons To Accept:**

- Succssful extension of previous deduction approaches to N-ary operators.
- Devise a hierarchical beam search procedure for the decoding procedure.
- Achieved state-of-the-art performance using less parameters.

**Reasons To Reject:**

- Paper organization: The explanation about CMWPA dataset is not sufficient. (I think the authors should bring some details about the dataset from the appendix).

**Reproducibility:**

4: Could mostly reproduce the results, but there may be some variation because of sample variance or minor variations in their interpretation of the protocol or method.

**Reviewer Confidence:**

4: Quite sure. I tried to check the important points carefully. It's unlikely, though conceivable, that I missed something that should affect my ratings.

**Typos Grammar Style And Presentation Improvements:**

- The introduction about CWMPA should appear, when you use the term CWMPA (line 434). The introduction about CWMPA is stated in lines 477-485, which is very far from line 434.
- Could you add the number of parameters for each model, in Table 1?

---

> ### Author Rebuttal · Authors · 2023-08-26
>
> Question A:
> In fact, we used multi-head attention but with **num_attention_heads**=1  in our experiments. This hyperparameter was chosen through validation set analysis, revealing that the model's optimal performance is achieved when **num_attention_heads**=1.  we will revise our paper to ensure consistency.
>
> Question B:
> We appreciate your suggestions, and we will transfer the primary description of the CWMPA dataset to the main paper.

---

### Meta-Review · Area_Chair_vJcw · 2023-09-15

**Recommendation:** 5

**Metareview:**

The reviewers highlight the novelty and exciting contributions of this paper. The paper is of good quality, both in terms of writing and analyses. There are some reasons to reject the paper, but they are mainly concerned with adding some clarifications or restructuring the paper.

Paper Topic And Main Contributions:

This paper introduces a novel deductive approach called GeDe for solving math word problems (MWP) by generating operations iteratively using a multivariate directed acyclic graph (mDAG) structure. GeDe addresses issues related to repeated subtree generation and binary tree expression limitations. It achieves this by re-encoding the input at each reasoning step, introducing new intermediate quantities, and decoding to generate multivariate operations. The paper also presents a hierarchical beam search to enhance model performance. Experimental results on multiple datasets, including the newly released CMWPA dataset, demonstrate GeDe's effectiveness in achieving state-of-the-art performance with fewer parameters while properly handling N-ary operators.

Reasons To Accept:

* Novelty of the Proposed Approach
    * Successful extension of previous deduction approaches to N-ary operators, addressing a critical limitation in math word problem-solving.
    * Devising a hierarchical beam search procedure for the decoding process, enhancing the model's efficiency and performance.
    * Achievement of state-of-the-art performance while using fewer parameters, demonstrating the paper's contribution to optimization in the field.
* New Data Set
    * Collection and utilization of a dataset to showcase the effectiveness of the proposed GeDe approach, including dynamic quantity embeddings, re-encoder, and multivariate operation mechanisms.
* Quality of the Writing
    * Clear and well-structured writing
* Quality of the Analyses
    * Insightful ablation analysis.
    * Promising results and insights from related experiments can potentially inform and benefit future research in the field of math word problem-solving.

Reasons To Reject:

* The core of the paper lacks some details that should be included in there and not in the appendix.
* The authors did not report the setting of random seeds or the average performance during multiple experiments.
* The authors need to add more explanations of the results.

---

### Decision · Program_Chairs · 2023-10-07

**Decision:**

Accept-Main

**Comment:**

The reviewers highlight the novelty and exciting contributions of this paper. The paper is of good quality, both in terms of writing and analyses. There are some reasons to reject the paper, but they are mainly concerned with adding some clarifications or restructuring the paper.

Paper Topic And Main Contributions:

This paper introduces a novel deductive approach called GeDe for solving math word problems (MWP) by generating operations iteratively using a multivariate directed acyclic graph (mDAG) structure. GeDe addresses issues related to repeated subtree generation and binary tree expression limitations. It achieves this by re-encoding the input at each reasoning step, introducing new intermediate quantities, and decoding to generate multivariate operations. The paper also presents a hierarchical beam search to enhance model performance. Experimental results on multiple datasets, including the newly released CMWPA dataset, demonstrate GeDe's effectiveness in achieving state-of-the-art performance with fewer parameters while properly handling N-ary operators.

Reasons To Accept:

* Novelty of the Proposed Approach
    * Successful extension of previous deduction approaches to N-ary operators, addressing a critical limitation in math word problem-solving.
    * Devising a hierarchical beam search procedure for the decoding process, enhancing the model's efficiency and performance.
    * Achievement of state-of-the-art performance while using fewer parameters, demonstrating the paper's contribution to optimization in the field.
* New Data Set
    * Collection and utilization of a dataset to showcase the effectiveness of the proposed GeDe approach, including dynamic quantity embeddings, re-encoder, and multivariate operation mechanisms.
* Quality of the Writing
    * Clear and well-structured writing
* Quality of the Analyses
    * Insightful ablation analysis.
    * Promising results and insights from related experiments can potentially inform and benefit future research in the field of math word problem-solving.

Reasons To Reject:

* The core of the paper lacks some details that should be included in there and not in the appendix.
* The authors did not report the setting of random seeds or the average performance during multiple experiments.
* The authors need to add more explanations of the results.